# Does Corporate Financialization Have a Non-Linear Impact on Sustainable Total Factor Productivity? Perspectives of Cash Holdings and Technical Innovation

Hui Wang [1,*], Qing Wang [2] and Xia Sheng [2]

1. School of Business Administration, Southwestern University of Finance and Economics, Chengdu 610031, China
2. Institute of Chinese Financial Studies, Southwestern University of Finance and Economics, Chengdu 610031, China; wqing@swufe.edu.cn (Q.W.); l_leigh@126.com (X.S.)
* Correspondence: daisy8611@gmail.com

**Abstract:** This study explores the conditions under which financialization may foster sustainable total factor productivity (TFP). We examine the inverted U-shaped relationship between corporate financialization and TFP by employing a panel threshold model using microeconomic non-financial panel data from Chinese firms in the 2007 to 2018 period. Our results suggest that the turning point is more significant in holding short-term financial assets and state-owned enterprises. The threshold effect suggests that technical innovation determines the optimal threshold at which TFP is affected by financialization. Further, financialization is considered an alternative to cash in order to increase the value of capital, leading to a positive effect on TFP. Contrary to their positive effects below the optimal thresholds, financialization exceeds a certain level, displaces technical innovation, and becomes detrimental to TFP. Our analysis thus establishes the importance of sustainable growth of TFP and minimize the adverse effect of financialization.

**Keywords:** circular economy; sustainability strategy; resilience; financialization; TFP; innovation

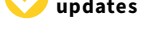



## 1. Introduction

Following the widespread outbreak of Covid-19, China's the real economy is headed for downturn due to the low production efficiency. As the entity economy gradually shifts more from industrial sector to financial sector, productivity growth has been slowed down and there have been concerns about economic resilience. The need for strengthen the resilience related to the creation of sustainable growth is clearly identified by D'Adamo and Rosa [1]. One aspect of economic resilience that is often underappreciated concerns the sustainable productivity growth and the imbalance of reallocate resources. Notably, financial assets are a vital component of the capital composition, which have a significant impact on the aggregate productivity growth. As the pandemic spreads, given rapidly changing economic, competitive, and consumer trends. These trends presents the adoption of circular economy principles associated with sustainability have become relatively more significant [2]. A circular strategy gradually becomes a prime concern of corporate executives and policymakers [3]. Financialization is considered a key feature for capital extension and to reach higher production efficiency yields. Nevertheless, Over-financialization in economy may lead to a downward trend of the entity sector, yet that cannot be confirmed at this time. This paper aims to test this hypothesis by analyzing the potential nonlinear relationship between corporate financialization and TFP growth. Our analysis establishes the importance of sustained growth of TFP and minimize the adverse effect of financialization. More importantly, technologies, policies, and financial activities must consider the sustainability aspect [2].

China's economy growth has declined steadily in recent years. In fact, the decline in TFP explains most of the fall in economy growth since the global financial crisis exploded

in 2008. In response, Chinese government has pushed expansionary fiscal and monetary policy to stimulate domestic economy growth while, stimulus policies have laid the key foundation for the increased development of financial market. The proportion of financial assets within the Chinese economy has skyrocketed, and the preference for corporate governance to hold financial assets continues to spread. A large share of financial investments is concentrated in firm portfolios, which grew from \$39.3 trillion to \$141.2 trillion from 2007 to 2018.

Growth in the financial development of non-financial corporations has received considerable attention in recent years. Several researchers and institutions have defined financialization as the proportion of financial assets held by non-financial corporations. It mainly implies that the financial assets contained in firm portfolios become a primary component of the capital expenditures highlighted in the analyses of the financialization trend [4]. On the other hand, since the 2008 United States subprime mortgage crisis, China's demographic dividend gradually disappeared, and the direct contribution of labor to gross domestic product (GDP) began to decline. The resource reallocation that China gained through the intersectoral transfer of labor and stable capital-return that resulted from the unlimited supply of labor will gradually disappear. For a long time, the catalyst of China's rapid GDP growth depended on the growth of total factor productivity (TFP), which means additional economic output is produced from a given amount of inputs. The driving forces of TFP have changed as capital plays a central role in the sustained economic growth of TFP. China's economy has promoted its shift in focus from high-speed growth to high-quality development in the midst of economic financialization, which means it has transitioned from factor-driven and investment-scale driven development to innovation-driven development. Meanwhile, the mutual integration of production efficiency, resource allocation efficiency, and technological innovation is particularly important. Therefore, the consequences of the corporate financialization for TFP are extensive.

There is an ambiguous relationship between financial development and TFP. The current studies on corporate financialization and TFP have not been agreed upon. Corporate financialization in terms of investment behavior is demonstrated by non-financial firms' frequent participation in financial markets and financial transactions [5]. It is also demonstrated by an increase in the contribution of financial gains to total corporate profits [6]. Studies have shown that firms hold financial assets as an expedient way to hedge liquidity risk [7–9] and reduce the risk of liquidity crunch by obtaining returns on financial investments [10], which can safeguard production and operations and contribute to high-quality development.

Based on summarizing existing research conclusions, we explore the impact of TFP on increased financial investments by non-financial firms and detect the dual mechanism through the following channels: (1) For cash holdings, in general, the level of cash holdings is considered an important basis for the allocation of production inputs and capital. Financial assets with low conversion costs not only manage liquidity shocks, but their excess returns can optimize profits and smooth the capital requirements of firms [11], which can effectively drive TFP growth from the accumulation of capital. (2) For technical innovation, research and development (R&D) investments play a key role in promoting the innovation initiative. The technology spillovers through openness are beneficial for TFP growth.

Unlike other works, standard regression models assuming a linear relationship between the two variables may have led to biased and misleading results. Our main assumption is that the distinct economic effects of financialization depend on the motives of holding financial assets and the degree of financialization. The objective of this paper is to examine whether the relationship between financialization and TFP is non-linear using samples of Chinese listed nonfinancial companies during the period from 2007 to 2018. By applying the threshold model introduced by Hansen [12], our empirical results confirm that the nexus between financialization and TFP is indeed non-linear. It shows that there exists a single significant threshold value of 0.13 above which the negative impact of financialization being found on TFP. The threshold model also allows us to measure the

respective roles of cash holding and innovation. We therefore find a statistically significant threshold effect from the view of innovation, and cash holding only partly supported in the relationship between financialization and firms' TFP. Our empirical results indicate robust results that financialization does not always lead to TFP growth. Accordingly, corporate governance should undertake a substantial investment plan to generate sustained growth of TFP and minimize the adverse effect of financialization. The findings obtained from this research may offer meaningful policy implications and additional knowledge to this growing literature on financialization.

This study explores the conditions under which financialization may result in improved TFP, which is seldom discussed in previous literature. There are three major contributions. First, this study has the advantage of detecting potential nonlinearity in the relationship between financialization and TFP. In this sense, our paper contributes to the understanding of the impact of corporate financialization. Moreover, this study uses the panel threshold methodology to verify the dual effects of the financialization, where the threshold effect of financialization on TFP would differ above and below this level. To our knowledge, there are no published empirical studies that reveal the underlying mechanism by applying the threshold effects of cash holdings and the innovation initiative. The empirical analysis does emphasize the importance of innovation in determining the relationship between financialization and TFP. According to this finding, the nonfinancial corporates should review their allocation of production factors from the perspective of productivity, and pay keen attention to enhance their resilience in the face of shifting economic conditions. Furthermore, there is not enough study exploring the heterogeneous features of financialization. Our paper examined the various consequence of financialization based on the structures of financial assets and corporate ownership. Lastly, we reference the new capital management regulations issued in 2017 to address excessive investments in financial products as a quasi-natural experiment leading to a more extensive study.

The remainder of the paper is organized as follows. Section 2 presents a review of the relevant literature, emphasizing nonlinearities in the corporate financialization nexus. Section 3 is research design, which describes the samples, variables, and models. Section 4 presents empirical findings and detailed discussion, including baseline regression analysis, heterogeneity analysis, threshold regression analysis, and robustness test. Whereas the conclusion and summary of the findings are discussed in the final section.

## 2. Literature Review and Research Hypothesis

### 2.1. The Impact of Financialization on Economy

There is extensive theoretical literature on the impact of financialization on the economy from macroeconomic perspectives. Most economists generally view excessive financialization as a significant obstacle for economic development [4,13–15]. Several previous studies [16–18] empirically establish that excessive financialization has a negative effect on capital accumulation when resource and production input factors are established. The findings of Singh [19], Krugman and Anthony [20] and Orhangazi [4] indicate that financialization hinders economic growth by extracting additional profits from the economy into the financial sector; likewise, corporate expenditures are allocated from production activities to financial investments. Sweezy [21] argues that the dramatic expansion of the financial sector, high degree of independence within the financial sector, and gradual dominance of the real production system pose potential financial risks to the economy. The findings of Lazonick [22] demonstrate that the overspending of manufacturing firms leads to decreased investments in production and increased unemployment rates. The empirical result implies that over-financialization has a negative effect on unemployment. China's increasing capital flows into the stock market and real estate industries represent expanding financialization for the economy. Zhang et al. [23] examines the negative impact of excessive spending on the enterprise's physical investment ratio and indicates that these aspects play a significant role in weakening monetary policy. Wang et al. [24] also confirms that excessive corporate financialization exacerbates asset bubbles. In particular, Law and

Singh [25] explore the possible asymmetric relationship between the extension of financial resources and growth, which indicates that the expansion of the financial system benefits growth to a certain extent. Sahay et al. [26] presents a similar argument by indicating that excessive financial resources increases economic risk and financial volatility.

### 2.2. The Impact of Corporate Financialization

Previous papers have emphasized that corporate investment behavior is associated with financialization. It is important to summarize the main findings of some influential studies. The increase in financial returns is related to a decrease in industrial returns, and corporate financialization makes non-financial firms hold less capital [27,28]. This indicates that high profits through financial channels are strongly associated with lower investments and capital accumulation. Significantly high financial assets are speculative and opportunities for operational growth are ignored. Krippner [29] studied the effects of financialization in the United States from 1950–2000 and found a negative correlation between manufacturing and the financial composition of corporate profits. Based on the historic performance of the Chinese economy, a recent study by Zhang and Zhang [30] found that pursuing high profits is one of the main reasons for corporate financialization at the expense of production inputs. High profits have ended investments in the economy. Investments resulting from increased profits might decreased operational spending in the manufacturing industry, which requires long term investment cycles and poses uncertainties in technology and risks. To maintain a stable profit, corporate managers seek opportunities to adjust their balance sheets and increase revenue from capital investments. The majority of the manufacturing industry has transitioned from traditional production activities to financial channels [6,31]. Since this transition, corporate financialization is defined as the increase in profits from unproductive business activities. In this case, corporate managers seek capital appreciation rather than operating profits. A significant portion of revenue is composed of profits [32], which is an example of corporate financialization.

### 2.3. Nonlinear Effects of Financialization on Entrepreneurship

There are two types of literature that cover the financial assets of firms and capital accumulation. Two main views exist, leading to ambiguous conclusions. One group of researchers believes that moderate financialization results in high capital gains and secures TFP. The second group of researchers believe that excessive financialization hinders the growth of TFP by inhibiting technological innovation and capital accumulation. However, both of these arguments are true to some extent. Thus, the extent to which TFP is affected depends on why financial assets are held. If motivated by speculation, firms will hold more financial assets considering the balance between risk and return.

When there has been a change in corporate management or corporate management is unable to meet the company's financial goals, they allocate financial assets [10,33]. This allocation of financial assets enhances capital liquidity, improves financing capacity, increases the return on assets [7,9] increases short-term shareholder value, and integrates production and finance [34], which contributes to the growth of TFP. Non-financial firms can capture higher returns during market booms by investing in diversified financial instruments, which provides a cushion and reduces risk during market downturns [8]. Arcand et al. [35] argue that the essential function of finance is to serve the entire economy. Only when financial development exceeds reasonable limits, does it shift from promoting economic growth to inhibiting economic growth. Again, as the return on investment of financial assets is higher than that of physical investments, enterprises will rely on financial investment income rather than working to improve operational efficiency. Liquidity plays a crucial role in investment decisions [36]. Financialization offers firms the flexible option of investing in reversible short-term financial assets instead of irreversible long-term physical assets; therefore, financial assets displace productivity accumulation as preferred by shareholders [37] mainly as a result of change in corporate management [22,38]. Therefore, this management leads to changes in decision-making related to capital structure and production alloca-

tion. This evidence indicates that a reasonable level of financialization does not hinder production, but it may effectively promote technology upgrades and improvements of research and development (R&D) [34]. Zhang and Luo [39] also argue that financialization of private firms contributes to the improvement of productivity improvement through actions such as reducing financing costs and easing financing constraints.

Bonfiglioli [7] identified that financialization could broaden finance options to allocate capital more efficiently. Adequate capital would give businesses and investors more choice and improve resource allocation; besides, financial constraints can limit the inputs in R&D, which is a significant determinant of TFP [40]. More specifically, financial support is provided for firms' technological progress, upgrade of human capital, and productivity improvements. Overall, moderate financialization has a profound impact on enhancing the ability to create value, but excessive financialization is likely to lead to the misallocation of productivity factors, which ultimately affects the productive efficiency.

The growth theory suggests that the original driver of economic growth is productivity [41]. Economists generally agree that technology leads to productivity improvement; in other words, the increase in the growth of TFP is driven by technological innovation [42]. More recently, Seo et al. [13] examined nonfinancial Korean corporations from 1994 to 2009 and found that increased financial investment and profit opportunities displaced R&D investment. Likewise, Xu and Liu [43] empirically examine the impact of financial asset allocations on R&D activities in China from 2007 to 2015, and the results show evidence of a strong negative correlation between financial asset allocations and firms' innovations. From the empirical point of view, financialization may affect firms' productivity through technological improvement. This paradigm is based on the realization that technological innovation is a long-term capital input that contributes to growth through the reallocation of productive resources. Economic outcomes are difficult to determine because corporate investment decisions are complex due to the trade-off between short-term profits that are not guaranteed and sustainability strategies. Orhangazi [4] explains that a higher return from financial activities should drive a change in the priorities of strategies. A reasonable level of financialization plays an active role in generating new profit sources, thus improving liquidity. However, excessive financialization that replaces long-term R&D investments with short-term profits that are not guaranteed may displace resources for economic development.

Obviously, the results derived from the above researches are not conclusive in matters of the exact relationship of financialization and TFP. Most of the empirical studies are based on ordinary least squares, which ignores the existence of asymmetries. Hence, based on the existing literature, we have assumed non-linearity in the relationship between financialization and TFP, to be empirically verified at a later stage of the study.

Following the arguments above, this study posits that holding excessive amounts of cash destroys the firm's value through maintenance costs. Therefore, in a comparable level of financialization, the effects of capital accumulation positively impact TFP. However, when the level of financialization exceeds a certain point, displacement, low resource allocation, and efficiency offset the positive effects of capital accumulation. This paper hypothesizes that the positive effects of financialization on innovation declines when a certain threshold is exceeded. Based on the above discussion, the paper hypothesizes the following:

**Hypothesis (H1).** *The relationship between financialization and TFP is asymmetric.*

*The level of cash holding is not significant determinants of TFP in this sample because of coexistence of what I call "positive" and "crowding out channels" of effect running from technical innovation to factors that affect TFP in the growth prospects. Evidence of the existence of both types of channel will be presented.*

**Hypothesis (H2).** *The threshold effect between financialization and TFP depends on a certain level of cash holdings and innovation.*

As an alternative method, we propose a quadratic explanatory variable to examine the dual effect. To further analyze the channels through which financialization affects TFP, this paper applies the Hansen [12] fixed effect panel threshold model convey to explain non-line relationships.

## 3. Empirical Methodology and Research Design

### 3.1. Basic Model Specification and Threshold Model Construction

As aforementioned, the impact of financialization on TFP is not necessarily a simple linear relationship. To allow for nonlinearity in the relationship, we also include the quadratic term of the financialization ($Fin_{i,t}{}^2$) in the model to examine test the research hypothesis proposed above:

$$TFP_{i,t} = \alpha_0 + \alpha_1\, Fin_{i,t} + \alpha_2\, Fin_{i,t}{}^2 + \alpha_3 \sum Control_{i,t} + \mu_i + \sigma_t + \varepsilon_{i,t} \tag{1}$$

where $TFP_{i,t}$ stands for the nonfinancial firms total factor productivity variable, for the measurement, it is important to notice that each of different estimates TFP measures may be affected by important statistical issues and limitations. This paper adopted the approach of Olley and Pakes [44]. $Fin_{i,t}$ indicates the level of financialization; The variable $Control_{i,t}$ is a vector of control variables, we include several factors that potentially affect the level of TFP. The $i$ and $t$ indicate cross-section (nonfinancial firms) and time period (2007–2018), respectively.

To verify underlying mechanism, we apply the threshold regression model introduced by Hansen [12], which is widely used in economics. This threshold model allows to split the effects of a key independent variable on dependent variable into regimes based on the value of a threshold, which can be expressed as follows:

$$TFP_{i,t} = \gamma_0 + \gamma_1\, CI_{i,t}\mathrm{I}(\, Fin_{i,t} \le \lambda) + \gamma_2 CI_{i,t}\mathrm{I}(\, Fin_{i,t} > \lambda) + \gamma 3 \sum Control_{i,t} + \mu_i + \sigma_t + \varepsilon_{i,t} \tag{2}$$

where $\mathrm{I}(\cdot)$ is the indicator function representing the sample splitting. The above regression model describes the sample split by only one threshold level. Whereas the parameter $\lambda$ is the threshold value, which assumed unknown and needs to be estimated. Here we select the level of financialization as the threshold variable.

The panel threshold model is well suited for testing the possible nonlinearity between financialization and TFP for two reasons. Firstly, as illustrated in Equation (2), since the sample is endogenously split according to a threshold value, the sign and the magnitude of the key variable are separately determined by two different subsamples. This procedure thus permits a flexible way in modeling potential nonlinear relationship between two variables. Secondly, the threshold parameter is estimated simultaneously along with other parameters, this means the estimated nonlinear pattern is discovered by optimally fitting the underlying data features, which minimizes specification concerns.

This paper prefers to explore the response of cash holdings and R&D investment channels to both appreciation and depreciation in the TFP, according to which we could then test the hypothesis to infer whether firm's financialization behavior is motivated by precautionary cash holdings or the crowding out of R&D investment mechanism.

To identify the attributes of virtual and non-virtual Chinese real estate during periods of sustained housing price increases, we reveal investment restrictions for 16 cities in 2017. Comprehensive firm-level data and the restructure rule are used to determine the endogenous issue.

### 3.2. Data and Empirical Strategy

The financial data refer to China listed nonfinancial companies taken from CSMAR database, which contains standardized accounting information about not only investment, sales, profits, interest and dividend payments but also types of financial assets. The initial number of firms includes 3654 firms for the period 2007–2018. As for control of the financial development, we use data from CCER database. We exclude the sample of companies with missing main variables, Special treatment (ST)/*ST firms, and select firms that have at

least three consecutive observations for the dependent variable, which is also required for econometric purposes we drop all the companies with a permanent negative total assets, an asset-liability ratio greater than one and negative owner's equity. Exclude Data anomalies and missing from such companies may affect the reliability of the results of this study. Finally, we exclude observations in the upper and lower 1% of each variable's distribution.

*Interpreted variable: total factor productivity (TFP)*. This paper measures the TFP uses the semi-parametric approach which is initiated from Olley and Pakes [44]. Specifically, following by Xiao and Xue [45], the firm's current investment is proxied by the net cash for acquisition, construction of fixed assets, intangible assets, and other long-term assets.

*Explanatory variable: Financialization.* The absolute levels liquid financial holdings remain vast. Financialization is characterized by the expansion of financial assets relative to entity activity of nonfinancial firms [46]. This paper uses the ratio of financial asset to the total assets, reflecting the level of financialization. Following main categories of financial assets are identified: (1) trading financial assets, (2) available-for-sale financial assets, (3) held-to-maturity investments, (4) investment properties, (5) derivative financial instruments, (6) long-term equity investments. These assets are probably highly liquid and easy convertible, while the cash held by the company is excluded as the motives typically for operational reserve rather than speculative investment purposes.

*Control variables*: This paper selects the following variables to control the firm-level and macro-level factors that may affect the TFP, including firm size (Size), profitability (Roa), growth (Growth), and financial leverage (Lev); macro-level factors include financial deepening (M2/GDP). In Table 1, the data descriptions are given.

**Table 1.** Data description.

| Variable | Definition |
|---|---|
| TFP | OP method |
| Fin | (Trading financial assets + net held-to-maturity investments + bought-back financial assets + available-for-sale financial assets + derivative financial assets + investment properties)/Total assets |
| Size | Logarithm of total assets |
| Cashflow | Logarithm of net cash flows from operating activities |
| Roe | Net profit/total assets |
| Age | Current year -year of establishment of each company |
| Growth | Annual growth rate of operating income |
| Tng | Fixed assets/total assets |
| Ltv | Total liabilities/total assets |
| Top | Number of shares held by the largest shareholder/total share capital |
| Capital | Net expenditure on acquisition and disposal of fixed assets, intangible assets and other long-term assets/total assets |
| Cir | Total assets/operating income |
| RD | R&D investment/total assets |
| M2/Gdp | M2/GDP |

## 4. Results and Discussion

Table 2 shows descriptive and normality statistics (mean, standard deviation, minimum, median, and maximum) of all variables of the study. The maximum value of Fin for nonfinancial enterprises in China is 43.01, the minimum value is $2.71 \times 10^{-10}$, and the standard deviation is 0.962, indicating that there are obvious differences in the level of financialization.

**Table 2.** Descriptive statistics.

| Variable | Obs | Mean | Std. Dev. | Min | Median | Max |
|---|---|---|---|---|---|---|
| TFP | 10243 | 19.210 | 1.363 | 10.909 | 19.203 | 24.490 |
| Fin | 10243 | 0.138 | 0.962 | $2.71 \times 10^{-10}$ | 0.0284 | 43.010 |
| Capital | 10243 | 0.0580 | 0.134 | $-0.253$ | 0.029 | 3.257 |
| Cashflow | 10243 | 18.270 | 1.929 | 7.409 | 18.415 | 25.396 |
| Size | 10243 | 21.090 | 1.239 | 13.680 | 21.770 | 28.060 |
| Ltv | 10243 | 0.397 | 0.205 | 0.007 | 0.385 | 0.984 |
| Roe | 10243 | 0.072 | 0.158 | $-6.797$ | 0.072 | 1.615 |
| Age | 10243 | 15.59 | 6.223 | 1 | 15 | 69 |
| Tng | 10243 | 1.426 | 1.972 | 0.007 | 0.798 | 19.96 |
| Top | 10243 | 33.656 | 14.722 | 3.00 | 31.050 | 89.99 |
| Tobinq | 10243 | 2.210 | 1.908 | 0.152 | 1.732 | 58.59 |
| Growth | 10243 | 0.171 | 0.376 | $-0.988$ | 0.121 | 4.792 |
| Cir | 10243 | 9.829 | 29.665 | 0.017 | 2.261 | 392.90 |
| Rd | 10243 | 0.055 | 0.361 | 0 | 0.014 | 11.11 |
| M2/Gdp | 10243 | 171.50 | 18.845 | 130.890 | 175.200 | 193.02 |

As shown in Table 3, the absolute value of the correlation coefficient of the main variables is less than 0.5, and the variance inflation factor VIF is less than 10, indicating that there is basically no multicollinearity among variables, and the selection of each variable is reasonable.

**Table 3.** Correlation matrix of variables.

| | Fin | Capital | Cashflow | Size | Ltv | Roe | Age | Tng | Top | Tobinq | Growth | Cir | Rd |
|---|---|---|---|---|---|---|---|---|---|---|---|---|---|
| Fin | 1 | | | | | | | | | | | | |
| Capital | $-0.004$ | 1 | | | | | | | | | | | |
| Cashflow | $-0.021$ | $-0.058$ | 1 | | | | | | | | | | |
| Size | $-0.058$ | $-0.212$ | 0.300 | 1 | | | | | | | | | |
| Ltv | 0.011 | $-0.059$ | 0.198 | 0.088 | 1 | | | | | | | | |
| Roe | 0.002 | 0.022 | 0.027 | $-0.063$ | $-0.048$ | 1 | | | | | | | |
| Age | $-0.014$ | $-0.102$ | 0.272 | 0.308 | 0.125 | $-0.070$ | 1 | | | | | | |
| Tng | 0.001 | $-0.015$ | 0.029 | 0.029 | 0.168 | $-0.025$ | 0.032 | 1 | | | | | |
| Top | 0.038 | 0.014 | 0.005 | 0.064 | 0.044 | 0.017 | $-0.029$ | 0.010 | 1 | | | | |
| Tobinq | $-0.001$ | $-0.010$ | $-0.001$ | $-0.049$ | 0.001 | 0.003 | 0.011 | $-0.002$ | $-0.039$ | 1 | | | |
| Growth | 0.007 | 0.020 | $-0.006$ | $-0.076$ | 0.010 | 0.146 | $-0.061$ | $-0.007$ | $-0.020$ | 0.020 | 1 | | |
| Cir | $-0.005$ | $-0.086$ | $-0.051$ | 0.319 | $-0.021$ | $-0.021$ | 0.017 | 0.001 | 0.034 | $-0.017$ | $-0.029$ | 1 | |
| Rd | 0.000 | 0.010 | $-0.041$ | $-0.094$ | 0.011 | 0.008 | $-0.028$ | 0.001 | 0.003 | 0.005 | 0.007 | $-0.009$ | 1 |

*4.1. Basic Regression Results*

Table 4 presents the results of specification, Column (1) focusing on only explanatory variables, as expected, the correlation between Fin and Fin2 are opposite statistically significant at the 1% statistical level. Column (2) shows that this dual relationship holds true after controlling for variables found to be important to TFP, such as firm characteristics, as well as controlling for industry characteristics. The results indicating that Fin has a positive impact, while Fin2 has a negative impact on TFP, which illustrates that an inverted U-shaped relationship between financialization and TFP. Columns (3) to (4), we include industry, year and city fixed effects respectively to control both heterogeneity in observable and unobservable characteristics and again find similar results. Column (4) illustrates the effects of the control variables, the nonlinear effects of Fin on TFP have a significance of 0.101 at the 1% statistical level, and those of Fin2 on TFP have a significance of $-0.002$ at the 5% statistical level.

**Table 4.** The U-shaped relationship between financialization and TFP.

| TFP | (1) | (2) | (3) | (4) |
|---|---|---|---|---|
| Fin | 0.096 *** | 0.101 *** | 0.099 *** | 0.101 *** |
| | (0.029) | (0.031) | (0.029) | (0.025) |
| Fin2 | −0.002 *** | −0.002 ** | −0.002 ** | −0.002 ** |
| | (0.001) | (0.001) | (0.001) | (0.001) |
| Capital | | 0.142 | 0.09 | 0.083 |
| | | (0.143) | (0.138) | (0.154) |
| Cashflow | | 0.328 *** | 0.327 *** | 0.308 *** |
| | | (0.013) | (0.011) | (0.012) |
| Size | | 0.263 *** | 0.264 *** | 0.260 *** |
| | | (0.018) | (0.018) | (0.02) |
| Ltv | | 0.695 *** | 0.713 *** | 0.660 *** |
| | | (0.129) | (0.129) | (0.14) |
| Roe | | 0.883 *** | 0.910 *** | 0.863 *** |
| | | (0.076) | (0.069) | (0.061) |
| Age | | 0.004 | 0.001 | 0.006 |
| | | (0.004) | (0.004) | (0.005) |
| Tng | | 0.000 | 0.000 | 0.000 |
| | | (0.001) | (0.001) | (0.001) |
| Top | | 0.001 | 0.001 | 0.001 |
| | | (0.007) | (0.001) | (0.001) |
| Tobinq | | 0.011 | 0.024 ** | 0.030 *** |
| | | (0.008) | (0.010) | (0.007) |
| Growth | | 0.077 *** | 0.051 *** | 0.060 *** |
| | | (0.017) | (0.018) | (0.015) |
| Cir | | −0.007 *** | −0.007 *** | −0.007 *** |
| | | (0.001) | (0.001) | (0.001) |
| Rd | | −0.042 | −0.062 | −0.082 |
| | | (0.102) | (0.093) | (0.081) |
| M2gdp | | 0.009 *** | - | - |
| | | (0.001) | - | - |
| Constant | 19.063 *** | 5.284 *** | 6.870 *** | 7.240 *** |
| | (0.044) | (0.427) | (0.546) | (0.541) |
| IndustryFE | No | Yes | Yes | Yes |
| YearFE | No | No | Yes | Yes |
| CityFE | No | No | No | Yes |
| Observations | 10,243 | 10,090 | 10,090 | 10,087 |
| R−squared | 0.002 | 0.32 | 0.329 | 0.389 |

Note: Standard errors in parentheses. ** $p < 0.05$, *** $p < 0.01$.

The coefficients of Fin are significant and positive, suggesting that an increase in financial assets level tends to improve TFP, but as financial assets proceeds further, it shows impediment. Moreover, the results of controlling variables are in line with the literature.

*4.2. Heterogeneity Studies*

The dual effect may vary for different types of financial assets and enterprises based on the asset structure and properties. Therefore, we explore heterogeneity by estimating separate regressions for the term structure of financial assets (Columns 1 and 2) and ownership structure of firms (Columns 3 and 4). This paper references Peng et al. [47] to categorize real estate and long-term equity investments as long-term financial assets based on the structure of financial assets, and the remaining category is short-term financial assets (total assets for standardized treatment). To study the effects of the financialization on different property rights enterprises, we categorize the samples according to the property rights and divide them into two sets of data: state-owned enterprises (SOE) and nonstate-owned enterprises (non-SOE).

Previous studies have found that liquidity is important to ensure that firms are able to meet short-term obligations [48] and meet the needs of daily business operations [49].

However, too much liquidity can be detrimental to profits. A financial asset is considered liquid if it can be converted into cash immediately or reasonably soon without a loss of value, also known as a cash equivalent. Therefore, good management of liquidity requires establishing a balance between cash holding and financial assets in order to maximize the firm's value and meet short-term obligations. The liquidity of financial assets varies with different time periods. As expected, the empirical results in Columns (1) of Table 5 show that the regression coefficient of the Fin and Fin2 in the sample of short-term financial assets is 0.046 and −0.012, respectively, which is significantly within the 1% confidence interval. The interaction between financialization and TFP turns out to be statistically significant only for short-term financial assets. Considering the short-term financial assets are characterized by high liquidity and low realization costs, enterprises have a stronger desire to seek financial profit through short term capital allocation and a higher degree of flexibility.

**Table 5.** Heterogeneity test.

| TFP | (1) | (2) | (3) | (4) |
|---|---|---|---|---|
| | Short-term financial assets | Long-term financial assets | SOE | Non-SOE |
| Fin | 0.046 *** | −0.176 | 0.037 ** | 0.044 |
| | (0.011) | (0.811) | (0.015) | (0.291) |
| Fin2 | −0.012 *** | 0.284 | −0.001 *** | 0.032 |
| | (0.0002) | (1.235) | (0.0002) | (0.053) |
| Controls | Yes | Yes | Yes | Yes |
| FirmFE | Yes | Yes | Yes | Yes |
| YearFE | Yes | Yes | Yes | Yes |
| Observations | 10,090 | 10,090 | 8386 | 1704 |
| R-squared | 0.269 | 0.147 | 0.302 | 0.288 |

Note: Standard errors in parentheses. ** $p < 0.05$, *** $p < 0.01$.

Financial constrains is generally thought to be closely related to investment behavior. In general, there is a financialization behavioral difference between SOE and non-SOE. In Columns (3) and (4), we find that financialization at SOE is sensitive to TFP, but the dual affection is not statistically significant for non-SOE, mainly because the availability of internal funds adds constraints to the investment decision. As noted, financial constraints play an important role in determining the optimal cash level and investment, directly impact investment decisions, and restrict production expansion, which impedes sustainable development and value maximization. For SOE, the degree of financing constraints is relatively low, and corporate management adjusts their management as needed. Corporate management depends on the principle of enterprise operations to maximize profits by optimizing input combinations. It seems that corporate governance may have incentives related to soft budget constraints to prefer the accumulation of financial assets over the creation of profit. On the other hand, large amounts of financing are channeled through SOE, which are much less efficient than China's private sector enterprises. The conclusion of existing studies shows that SOE are commonly perceived as performing poorly in TFP growth [50]. While SOE are unlikely to change their long-term production efficiency based on historical and policy factors, SOE could possibly gain a short-term profit. It implies that SOE have an advantage when involved in financial activities. Hence, they are prone to more financial assets when their low productivity revenue is adjusted due to the burden of excess capital. The increase in financial assets results from soft budget constraints, which is comparable to the fact that increased financialization in China is stronger in SOE. Compared to SOE, non-SOE have higher financial and budget constraints, which impose certain restrictions on capital use within the enterprises. Additionally, non-SOE are more likely to experience stagnant growth or bankruptcy due to financial constraints. However,

production and operation requirements ultimately determine strategies. Non-SOE tend to focus on material input and output instead of financial activity to drive productive growth.

*4.3. Fixed-Effect Panel Threshold Estimation Results*

To provide further insights into the non-linear relationship between financialization and TFP, we need to estimate the turning points. Table 6 column (1) shows that below the identified threshold $\lambda 1 = 0.0032$ the financialization do have a positive but statistically insignificant effect. However, we find a significant negative coefficient of the cash holding if the threshold value above 0.0032 which indicates that a high level of substitution of financial assets crowds out cash resources, thereby inhibiting TFP growth. Financialization through cash holding has an adverse effect on TFP, Thus, cash holding channel is only partly supported in the relationship between financialization and firms' TFP.

**Table 6.** Threshold regression.

| TFP | (1) | (2) |
|---|---|---|
| Fin | Threshold $\lambda 1 = 0.0032$ | Threshold $\lambda 2 = 0.13$ |
| Cashd0 | 0.022 (0.109) | |
| Cashd1 | −0.292 *** (0.078) | |
| In_ind0 | | 0.293 ** (0.148) |
| In_ind1 | | −0.365 *** (0.147) |
| FirmFE | Yes | Yes |
| YearFE | Yes | Yes |
| Observations | 7758 | 7758 |
| R-squared | 0.287 | 0.291 |

Note: Standard errors in parentheses. ** $p < 0.05$, *** $p < 0.01$.

It indicates that financialization does not improve cash value, however the decrease in cash holding is attributed to high level of financial assets. Evidence shows that firms holds more cash with a lower financial development market. In other words, the amount of cash that firms can hold is limited by financial assets. This reduces the operational efficiency of SOE. Furthermore, overspending by SOEs leads the state to control prices and tighten monetary policies, which reduces the productivity of the non-state sector and reduces economic growth.

Column (2) incorporates the results of the single threshold estimation. We find a single significant threshold value of 0.13 above which the relationship of the financialization and innovation turns nonlinear. It is important to note that up to a threshold of 0.13 the coefficient is 0.293, and above this threshold the coefficient declines slightly to −0.365. It illustrates that the turning point is 0.13, indicating that, before reaching this point, financialization has a positive relationship with TFP, while after this point, and the relationship becomes negative. From a theoretical perspective, the inverted U-shaped relationship shows that the impact of cash holding is bounded.

As discussed above, cash holdings have no significant effects on the firms' TFP, but innovation has an inverted U-shaped effect on TFP. Thus, it can be concluded that innovation has more significant effects on the firms' TFP. Innovation determines the growth of TFP in future sustainable development. This inconsistency can be attributed to the current stage of development in China, where policymakers use various policy instruments to promote the technological advances of firms. Particularly, it is important to note that most enterprises, especially private firms, face severe financial constraints in China. Therefore, these firms should rely on internal funding, which suggests that these firms are involved in activities that generate additional income. This can alleviate the pressure of external funding, which

will relieve financing constraints and reduce financing costs. This means that financial assets and the improved efficiency of additional income will result in financialization and capital value. Until recently, anecdotal evidence suggests that the participation of non-financial firms in financial markets can help enterprises obtain substantial financial support for technological innovation, and capital markets can support the progress of technological innovation by providing long-term incentive capital, risk diversification, and sharing opportunities for investors [51]. The advantage of holding financial assets, highlighted by Ang [52] and Arizala et al. [53], is that it enables firms to ease financial constraints and accumulate high income, which may contribute to technical innovation. Therefore, a reasonable level of financial assets is more likely to broaden the capital value and secure long-term innovation. A lower level of financialization is more likely to improve the retention of capital. When income is retained more sufficiently, enterprise managers will be more willing and able to seek long-term development of technology innovation instead of focusing only short-term benefits. This will provide financial support for the technological innovation of enterprises, promote the participation of enterprises in technological innovation activities, and have a positive impact on the efficiency of technological innovation. Better technology innovation is associated with higher TFP.

However, excessive financialization of enterprises will gradually separate the compensation of employees, especially managers, from their long-term performance and establish a relatively close relationship with the short-term arbitrage behavior in the financial market [54]. The spread of uncertainty has a negative impact on the corporate governance structure; furthermore, the substitution of investment funds can be used for non-R&D purposes, which results in the partial displacement of technological innovation.

The F-value and *p*-value obtained after 300 repeated samplings are presented in Table 7. The result shows that the single threshold effect of the model pass the test, it means a significant threshold effect of financialization exists, with the single threshold value of 0.13. Table 8 reports single threshold estimates and a 95% confidence interval. The LR value is less than 7.35, which is the critical value at 5% significant level. Figure 1 shows the estimation and confidence interval for single threshold.

**Table 7.** Test result of threshold significance.

| Threshold | F-Value | *p*-Value | 10% | 5% | 1% |
|-----------|---------|-----------|-----|-----|-----|
| Single | 17.43 ** | 0.017 | 8.831 | 12.617 | 19.145 |

Note: Standard errors in parentheses. ** $p < 0.05$, *** $p < 0.01$.

**Table 8.** Threshold values and confidence intervals.

| Model | Threshold Value | 95% Confidence Intervals |
|-------|----------------|--------------------------|
| Single threshold | 0.1300 | (0.0000, 0.4600) |

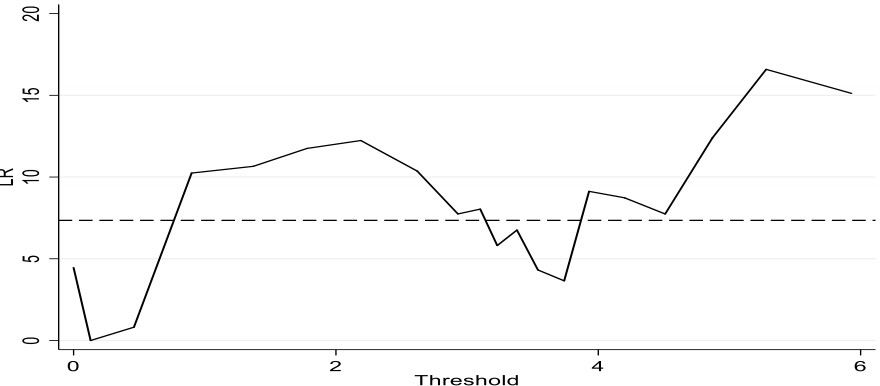

**Figure 1.** Estimation and confidence interval for single threshold.

*4.4. Robustness Test*

The possibility of endogeneity is an issue that might affect our study of the relationship between financialization and TFP. To overcome the potential issues of omitted variables and reverse causality among variables that may cause parameter estimates to become biased and inconsistent, we construct the following difference-in-differences model to conduct a more extensive test.

Specifically, we reveal the limited investment regulations of private equity funds resulting from the unexpected systemic risk enforced in 16 cities with elevated housing prices starting in 2017. This regulation may discourage these nonfinancial firms to invest in property. It is unlikely that the regulation has a direct effect on nonfinancial firms outside these 16 cities, allowing us to construct a control group to examine the heterogeneous effects across cities. A natural question is why the regulation is related to corporate financialization. Two facts about Chinese housing prices and real estate are well-documented and largely agreed upon. Official statistics in China show that housing prices grew dramatically between 2007 and 2014 [55] and moderately in recent years. This is indeed a real estate boom with Chinese characteristics, which is typically related to government decisions. As housing prices increased, relevant investments increased also. Compared to the downward trend of the entity sector, the profitability of the real estate sector is associated with a higher probability. Chinese owners or investors either directly purchase real estate or invest in the form of financial derivatives. According to the National Bureau of Statistics, China's total real estate investment was 0.36 trillion yuan in 1998 and increased to 10.98 trillion yuan in 2017, which rose nearly 30 times within 20 years. Existing data (China Wealth Management Product Market Development and Evaluation) shows that nearly 25% of trust funds flow to real estate, and housing price fluctuations lead to the conversion of properties from real asset attributes to financial attributes. Real estate also accounts for a major part of financial assets within Chinese nonfinancial firms. Since the late 2000s, the government has increasingly shifted its focus to financial stability and the imbalances between finance and the economy. Under the central government's guidance, regulators have sought to stabilize housing prices through restrictions and limiting investments in real estate financial assets.

In China, holding financial assets is a crucial form of real estate investments fueled by funding non-financial firms because they are flexible and highly liquid. The portfolios of financial assets are highly skewed towards real estate. Due to the fact that the boom and burst of real estate markets are closely related corporate investments [56], the difference-in-differences approach is used to compare the TFP before and after the regulation became effective. If that were the case, financialization would not be binding, which means that the constraint on holding real estate financial assets has a positive effect on TFP. Summarizing our empirical strategy, we estimate the following DID equation:

$$TFP_{i,t} = \lambda_0 + \lambda_1 \, Treat_i + \lambda_2 \, Time_t + \lambda_3 \, Did_{i,t} + \tau_k \sum Control_{i,t} + \mu_i + \sigma_t + \varepsilon_{i,t} \quad (3)$$

where $Treat_i$ is a dummy variable taking value of 1 if firm $i$ is in the 16 cities affected by the regulation. $Time_t$ takes value 1 if year is after 2017, and 0 otherwise. The regression controls for firm fixed effect, year fixed effects.

As it can be clearly seen from Table 9, once we include firm fixed effects, the absolute value of the coefficients decrease marginally, and hence accounting for the financialization quantitatively weakens the TFP, but economically the change is small. Whereas the results suggest that financialization was significantly affected by the regulation. This could potentially explain the increase in TFP reported in the period immediately after 2017 and why TFP was still negative before 2017, which could be driven by firms drawing on the level of financialization. Therefore, our main findings as above were confirmed.

**Table 9.** Difference-in-differences regression.

| TFP | (1) | (2) |
|---|---|---|
| Did | 0.107 ** | 0.059 * |
| | (0.055) | (0.036) |
| Controls | Yes | Yes |
| FirmFE | No | Yes |
| YearFE | No | Yes |
| Observations | 10,200 | 10,200 |
| R-squared | 0.30 | 0.268 |

Note: Standard errors in parentheses. * $p < 0.10$, ** $p < 0.05$.

It is worth discussing potential endogeneity concerns of our results. Firstly, all of our specifications, including the threshold regression model, have explicitly accounted for the individual fixed effects. These should eliminate endogenous bias caused by time invariant unobservable. Secondly, the remaining endogenous concern may come from reverse causality or simultaneously bias. Since the lagged explanatory variables tend to only be weakly correlated with current period's error in our main specification, we use lagged Fin to alleviate the potential endogeneity. Lastly, to overcome the potential bias caused by time-varying omitted variables, we re-estimated the quadratic regression and threshold regression by extensive control variables. Additional TFP determinant variables should be captured including industry and market characteristics. Specifically, Loan, the ratio of total loans to total debts, is used to control for the effect of lending capability. HHI, the Herfindahl-Hirschman index (HHI) as measured by the sum of the squared industry shares of each firm's assets for a given year, is used to control for the effects of industry concentration. Freturn, the financial return on investment is used to control for financial market return.

For the sake of brevity, we only report our main interest variables as shown in Tables 10 and 11, thus confirming the robustness of our previous findings.

**Table 10.** Lagged variables& Additional control variables- quadratic regression.

| TFP | (1) | (2) |
|---|---|---|
| | **Lagged Fin** | **Fin** |
| Fin | 0.050 * | 0.038 ** |
| | (0.029) | (0.015) |
| Fin2 | −0.002 * | −0.0016 *** |
| | (0.001) | (0.0004) |
| Loan | | −0.058 |
| | | (0.046) |
| HHI | | 0.039 |
| | | (0.079) |
| Freturn | | 0.337 *** |
| | | (0.081) |
| Controls | Yes | Yes |
| FirmFE | Yes | Yes |
| YearFE | Yes | Yes |
| Observations | 9306 | 9747 |
| R-squared | 0.267 | 0.27 |

Note: This table column (1) shows the lagged explanatory variables in quadratic regression, column (2) presents the regression with additional control variables. Standard errors in parentheses. * $p < 0.10$, ** $p < 0.05$, *** $p < 0.01$.

**Table 11.** Lagged variables& Additional control variables- threshold regression.

| TFP | (1) | (2) |
|---|---|---|
| | **Lagged Fin** | **Fin** |
| | Threshold λ1 = 0.116 | Threshold λ2 = 0.13 |
| In_ind0 | 0.213 * | |
| | (0.117) | |
| In_ind1 | −9.17 * | |
| | (0.529) | |
| Loan | | 0.083 |
| | | (0.057) |
| HHI | | 0.018 |
| | | (0.814) |
| Freturn | | 0.154 |
| | | (0.219) |
| FirmFE | Yes | Yes |
| YearFE | Yes | Yes |
| Observations | 5796 | 6680 |
| R-squared | 0.269 | 0.253 |

Note: This table column (1) shows the lagged explanatory variables in threshold regression, column (2) presents the regression with additional control variables. Standard errors in parentheses. * $p < 0.10$.

## 5. Conclusions

This paper presents new evidence regarding the effect of financialization on TFP in China. Our study contributes to the debate about the effect of holding financial assets of nonfinancial corporations, where no consensus emerges from prior literature. Our sample covered the period 2007–2018 for 3654 non-financial corporations, Through the use of a non-linear modeling strategy, we explore the relationship between financialization and TFP. Also we apply the threshold methodology to verify the mechanism between financialization and TFP from the perspectives of cash holdings and technological innovation. Through theoretical and empirical analysis, this paper concludes the following: Financialization, as measured by holding financial assets on total assets, has an inverted U-shaped relationship with the TFP. Taking heterogeneity into account, we find that the dual effect is more pronounced in short-term financial asset-holding and SOE. Furthermore, our findings indicate that there is a significant financialization threshold between technological innovation and TFP. In the low threshold interval (λ < 0.13), financialization can significantly promote TFP. Nevertheless, in the high threshold interval ((λ > 0.13). Since change of innovation explains the non-linear relationship better than cash holding. Such a threshold method adds perspective to existing models, which demonstrates the key role of innovation switches the effect of financialization. And in this case, further analysis is still needed to suggest a policy for limiting over financialization. Several extensions of our research would be desirable, including optimal productivity factor allocation, maximization of financial assets profit as well as the sustainable growth when face the shifting economic conditions.

Technological innovation and capital upgrades can fundamentally drive productivity growth and accelerate the circular economy. Our findings emphasize the role of financilization to determine a sustainable competitive advantage of capital as part of resilient and sustainable systems. The nonfinancial corporates should review their allocation of production factors from the perspective of productivity, and pay keen attention to enhance their resilience in the face of shifting economic conditions. Besides, a broader policy framework is required to promote the rational allocation of resources. To successfully manage economic volatility created by the pandemic and foster sustainable economic growth the policymakers need to implement joint actions to support the development of circular economy.



**Author Contributions:** Conceptualization, H.W. and Q.W.; methodology, H.W.; software, H.W.; validation, H.W., X.S.; formal analysis, H.W.; investigation, data curation, H.W., X.S.; writing—original draft preparation, H.W.; writing—review and editing, H.W.; supervision, Q.W.; project administration, Q.W.; funding acquisition, Q.W. All authors have read and agreed to the published version of the manuscript.

**Funding:** This research was funded by the National Natural Science Foundation of China [71950010].

**Data Availability Statement:** Not applicable.

**Conflicts of Interest:** The authors declare no conflict of interest.

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
