# Peer review of "Does Corporate Financialization Have a Non-Linear Impact on Sustainable Total Factor Productivity? Perspectives of Cash Holdings and Technical Innovation"

_sustainability, doi:10.3390/su13052533_

Round 1

Reviewer 1 Report

The article is of great interest, but needs to be improved on the following issues:
-
The methodology used must be adequately justified
-The limitations of the work and future lines of research are not raised
-It is convenient to indicate in the summary the methodology and sample used

Best regards,

Author Response

Dear Referee,

Thank you very much for evaluating and reviewing our manuscript. We are especially

grateful for the constructive comments and suggestions and we have to the best of our abilities responded to them. We address your comments in the following point by point response.

Comments:

1- The methodology used must be adequately justified paragraph

2- The limitations of the work and future lines of research are not raised

3- It is convenient to indicate in the summary the methodology and sample used

Response:

_ 1. We followed your advice to add a detailed description of the methodology. In addition, we also discuss the appropriateness of our method. (Page 13, line11)

The revised content is as follows:

To verify underlying mechanism, we apply the panel threshold regression model introduced by Hansen [9], which is widely used in economics. This threshold model allows to split the effects of a key independent variable on dependent variable into regimes based on the value of a threshold, which can be expressed as follows:

01I (≤λ)+γ2I(>λ)+γ3   (2)

Where I(∙) is the indicator function representing the sample splitting. The above regression model describes the sample split by only one threshold level. Whereas the parameter λ is the threshold value, which assumed unknown and needs to be estimated. Here we select the level of financialization as the threshold variable.

The panel threshold model is well suited for testing the possible nonlinearity between financialization and TFP for two reasons. Firstly, as illustrated in equation (2), since the sample is endogenously spitted according to a threshold value, so the sign and the magnitude of the key variable are separately determined by two different subsamples. This procedure thus permits a flexible way in modeling potential nonlinear relationship between two variables. Secondly, the threshold parameter is estimated simultaneously along with other parameters, this means the estimated nonlinear pattern is discovered by optimally fitting the underlying data features, which minimizes specification concerns.

_ 2. We have added the future research based on the limitations and findings of this paper in the Conclusion section. (Page 28, line10)

The revised content is as follows:

Since change of innovation explains the non-linear relationship better than cash holding.Such threshold method adds perspective to existing models, which demonstrates the key role of innovation switches the effect of financialization. And in this case, further analysis is still needed to suggest a policy for limiting over financialization. Several extensions of our research would be desirable, including optimal productivity factor allocation, maximization of financial assets profit as well as the sustainable growth when face the shifting economic conditions.

_ 3. We have added the description of methodology and sample used in the part of Abstract and the Introduction section. (Page1, line1 &Page5, line4)

The revised content is as follows:

The objective of this paper is to examine whether the relationship between financialization and TFP is non-linear using samples of Chinese listed nonfinancial companies during the period from 2007 to 2018. By applying the panel threshold model introduced by Hansen [9], our empirical results confirm that the nexus between financialization and TFP is indeed non-linear. It shows that there exists a single significant threshold value of 0.13 above which the negative impact of financialization being found on TFP. The threshold model also allows us to measure the respective roles of cash holding and innovation. We therefore find a statistically significant threshold effect from the view of innovation, and cash holding only partly supported in the relationship between financialization and firms’ TFP. Our empirical results indicate robust results that financialization does not always lead to TFP growth.

Reviewer 2 Report

This study is designed to verify potential impact of financialization on the economy from macroeconomic perspectives while specifically considering Chinese economical situations. That is, the relationship between financial development and TFP is not clearly and the authors tried to identify any possible non-linear relationship by applying the threshold effects of cash holdings and behavioral outcomes of the financialization of financial assets for heterogeneous firms.

In terms of overall organization of this study, this manuscript is categorized into four different sections based on statistical methods applied. This manuscript is finely confirmed with author submission guideline of this quality journal and the introduction part properly addressed relevant studies within underlying theoretical background.

References adopted in this study to support the authors’ theoretical opinions were considered to be reliable ones since those were published in quality referred journals.

The findings from this study would help potential audiences on the way of understanding TFP and financialization on economy status particularly for China.

Author Response

Dear Referee,

Thank you very much for evaluating and reviewing our manuscript, and we followed your advice and have now completely rewritten section of introduction and conclusion to make it more logical and clearer. Additionally, we have improved the writing of the paper throughout.

Once again, thank you very much for your constructive comments and suggestions which would help us to improve the quality of the paper.

Reviewer 3 Report

attached

Author Response

Dear Referee,

Thank you very much for evaluating and reviewing our manuscript. We are especially

grateful for the constructive comments and suggestions and we have to the best of our abilities responded to them. We address your comments in the following point by point response.

Comments:

1-The relation between financialization and productivity depends on different conditions. One important condition is the efficiency/maturity of the financial markets. You should clarify the contributions of the paper which are not elaborated well in the current paper. You can talk about the following contributions: What insights can you provide based on your finding? Do they push forward our understanding? What should we do with your research? Do you have any suggestions to improve the current regulation or practice? Adding the above discussion and extend your literature review may help you make more contributions and position your contributions better.

Response: Thank you very much for having raised these issues. We followed your advice and have now completely rewritten the contributions of the paper to make it clearer and more precise. (Page 5, line18)

The revised content is as follows:

This study explores the conditions under which financialization may result in improved TFP, which is seldom discussed in previous literature. There are three major contributions. First, this study has the advantage of detecting potential nonlinearity in the relationship between financialization and TFP. In this sense, our paper contributes to the understanding of the impact of corporate financialization. Moreover, this study uses the threshold methodology to verify the dual effects of the financialization, where the threshold effect of financialization on TFP would differ above and below this level .To our knowledge, there are no published empirical studies that reveal the underlying mechanism by applying the threshold effects of cash holdings and the innovation initiative. The empirical analysis does emphasize the importance of innovation in determining the relationship between financialization and TFP. According to this finding, the nonfinancial corporates should review their allocation of production factors from the perspective of productivity, and pay keen attention to enhance their resilience in the face of shifting economic conditions. Furthermore, there is not enough study exploring the heterogeneous features of financialization. Our paper examine the various consequence of financialization based on the structures of financial assets and corporate ownership. Lastly, we reference the new capital management regulations issued in 2017 to address excessive investments in financial products as a quasi-natural experiment leading to a more extensive study.

2- The paper seems to claim causality but does not discuss the potential endogeneity issue and its remedies sufficiently. The endogeneity problem can be driven by unobservable firm, manager, and market characteristics you need to discuss. In some sense, cash holdings, innovation, financialization are all endogenous choices that are simultaneously determined by the firms (i.e., shareholders and board of directors).

Response: Thank you for your comment, we add the discussion to the paper. (Page26, line9)

The revised content is as follows:

It is worth discussing potential endogeneity concerns of our results. Firstly, all of our specifications, including the threshold regression model, have explicitly accounted for individual fixed effects. These should eliminate endogenous bias caused by time invariant unobservable. Secondly, the remaining endogenous concern may come from reverse causality or simultaneously bias. Since the lagged explanatory variables tend to only be weakly correlated with current period’s error in our main specification, we use lagged Fin to alleviate the potential endogeneity. Lastly, to overcome the potential bias caused by time-varying omitted variables, we re-estimated the quadratic regression and threshold regression by extensive control variables. Additional TFP determinant variables should be captured including industry and market characteristics. Specifically, Loan, the ratio of total loans to total debts, is used to control for the effect of lending capability. HHI, the Herfindahl-Hirschman index (HHI) as measured by the sum of the squared industry shares of each firm’s assets for a given year, is used to control for the effects of industry concentration. Freturn, Financial return on investment is used to control for financial market return.

For the sake of brevity, we only report our main interest variables as shown in Tables 10-11, thus confirming the robustness of our previous findings.

Table10. Lagged variables& Additional control variables- quadratic regression

TFP

(1)

Lagged Fin

(2)

    Fin     

Fin

Fin2

0.050*

(0.029)

-0.002*

(0.001)

0.038**

(0.015)

-0.0016***

(0.0004)

Loan

HHI

Freturn

-0.058

(0.046

0.039

(0.079)

    0.337***

    (0.081)

Controls

FirmFE

YearFE

Observations

R-squared

Yes

Yes

Yes

9,306

0.267

Yes

Yes

Yes

9,747

0.270

Note: This table column (1) shows the lagged explanatory variables in quadratic regression, column (2) presents the regression with additional control variables. Standard errors in parentheses. *p <0.10, **p < 0.05, ***p < 0.01.

Table11. Lagged variables& Additional control variables- threshold regression

TFP

(1)

Lagged Fin

(2)

Fin

Threshold λ1=0.116

Thresholdλ2=0.13

In_ind0

In_ind1

Loan

HHI

0.213*

(0.117)

-9.17*

(0.529)

0.083

(0.057)

0.018

(0.814)

Freturn

0.154

 (0.219)

FirmFE

YearFE

Observations

R-squared

Yes

Yes

5,796

0.269

Yes

Yes

6,680

0.253

Note: This table column (1) shows the lagged explanatory variables in threshold regression, column (2) presents the regression with additional control variables. Standard errors in parentheses. *p <0.10, **p < 0.05, ***p < 0.01.

3- Try to avoid long sentences and vague words. Use short, precise, and concise sentences and be more straightforward. The last section of conclusion should summarize all your findings, their implications to researchers and practitioners, future direction for research, limitation of the current study, etc. You need to seriously proofread the paper and extend and update your references.

Response: Thank you very much for having raised this issue, we followed your advice and have now completely rewritten section of conclusion to make it more logical and clearer. Additionally, we have improved the writing of the paper throughout.

Round 2

Reviewer 1 Report

I congratulate the authors for the improvement of the article.

Author Response

Thanks.

Reviewer 3 Report

well done

Author Response

Thanks.